# Prospection of Psychrotrophic Filamentous Fungi Isolated from the High Andean Paramo Region of Northern Ecuador: Enzymatic Activity and Molecular Identification

**DOI:** 10.3390/microorganisms10020282

**Published:** 2022-01-26

**Authors:** Stefan Alexander Brück, Alex Graça Contato, Paul Gamboa-Trujillo, Tássio Brito de Oliveira, Mariana Cereia, Maria de Lourdes Teixeira de Moraes Polizeli

**Affiliations:** 1Facultad de Ciencias Biológicas, Universidad Central del Ecuador, Quito 170403, Ecuador; brueck.stefan@gmail.com (S.A.B.); jpgamboa@uce.edu.ec (P.G.-T.); 2Departamento de Bioquímica e Imunologia, Faculdade de Medicina de Ribeirão Preto, Universidade de São Paulo, Ribeirão Preto 14049-900, Brazil; alexgraca.contato@gmail.com; 3Facultad de Ingeniería Química, Universidad Central del Ecuador, Quito 170521, Ecuador; 4Departamento de Biologia, Faculdade de Filosofia, Ciências e Letras de Ribeirão Preto, Universidade de São Paulo, Ribeirão Preto 14050-901, Brazil; oliveiratb@yahoo.com.br (T.B.d.O.); macereia@ffclrp.usp.br (M.C.)

**Keywords:** psychrotrophic fungi, high Andean Paramo, cold-adapted enzymes, bioprospecting

## Abstract

The isolation of filamentous fungal strains from remote habitats with extreme climatic conditions has led to the discovery of a series of enzymes with attractive properties that can be useful in various industrial applications. Among these, cold-adapted enzymes from fungi with psychrotrophic lifestyles are valuable agents in industrial processes aiming towards energy reduction. Out of eight strains isolated from soil of the paramo highlands of Ecuador, three were selected for further experimentation and identified as *Cladosporium michoacanense*, *Cladosporium* sp. (cladosporioides complex), and *Didymella* sp., this last being reported for the first time in this area. The secretion of seven enzymes, namely, endoglucanase, exoglucanase, β-D-glucosidase, endo-1,4-β-xylanase, β-D-xylosidase, acid, and alkaline phosphatases, were analyzed under agitation and static conditions optimized for the growth period and incubation temperature. *Cladosporium* strains under agitation as well as incubation for 72 h mostly showed the substantial activation for endoglucanase reaching up to 4563 mU/mL and xylanase up to 3036 mU/mL. Meanwhile, other enzymatic levels varied enormously depending on growth and temperature. *Didymella* sp. showed the most robust activation at 8 °C for endoglucanase, β-D-glucosidase, and xylanase, indicating an interesting profile for applications such as bioremediation and wastewater treatment processes under cold climatic conditions.

## 1. Introduction

The High Andean mountain region is marked by the paramo ecosystem, neotropical grassland covering mountainsides from 3500 up to 5000 m of altitude from Southern Venezuela, Colombia, Ecuador, and Northern Peru [1]. It is considered a hotspot of biodiversity mainly due to the fact of its high degree of endemic species adapted to extreme environmental conditions [2] such as intense radiation close to the equator line [3], nighttime temperatures close to the freezing point, and low availability of nutrients [4]. Fungi growing under these conditions tend to develop a psychrophilic or psychrotrophic profile, developing at 0 °C with optimum growth conditions of ≤15 and 15–20 °C, respectively [5,6]. This restricted ecological group has been found in terrestrial and marine environments from Polar Regions, deep water and marine sediments of the oceans, and high mountains [7,8,9].

The Paramo Region of South America is a vastly unexplored area in terms of soil-derived microorganisms. Geospatial separation by Andean mountain chains creates a series of unique habitats, forcing microbes to adapt to a variety of given conditions, increasing the biological diversity with unique species within different paramo regions [10]. Most surveys of soil-borne fungi were conducted in the Colombian Paramo Region [11,12,13]. Information about fungi composition in Ecuadorian paramo soils is very scarce [14], and screenings for active enzymatic fungi with possible industrial applications have not been performed yet. Fungi isolated from the paramo can secrete cold-adapted enzymes [15], which are interesting in industrial procedures requiring low temperatures and employing energy to cool down the process. Nowadays, cold-active enzymes are mostly used in meat tenderization, food processing, flavoring, baking, brewing, cheese production, and animal feed [16,17]. They require low activation energies while showing during the meantime the highest activities, at low temperatures, of up to a 10-fold increase compared to mesophilic homologues [18], allowing for energy reduction [19], which might become increasingly important due to the tendency towards neutral carbon dioxide balances in times of global warming [20].

Enzymes can break down plant cell wall components, such as those of the cellulolytic system (endoglucanases, exoglucanases, and β-glucosidase), for efficient cellulose cleavage and of the xylanolytic system (mainly endoxylanase and β-xylosidase) for xylan breakdown to the level of reducing sugars. They are widely applied to biotechnological processes in areas such as the food industry, production of fuels, detergents, and biopulping [21,22,23,24]. Especially interesting for these cold-adapted enzymes are the treatment of wastewater and environmental bioremediation in countries with shallow temperatures due to the necessity of stable in situ applications without the need for heating [25,26].

Fungi further gain increasing importance as biofertilizers to improve crop yield. Given the broad range of climatic zones where crops are planted, biofertilizers must be equally adapted to these conditions including cold regions with generally lower productivity [6]. Cold adapted phosphatases can hydrolyze organic phosphate sources, which become assimilable by plant roots, improving their economic and ecologic growth, eliminating the need for chemical fertilizers [9,27].

The present study, therefore, aimed to investigate filamentous fungi from paramo soil and optimize incubation conditions for optimal enzyme activity yield to analyze whether their extreme living conditions led to adapted enzymes of the carbon and phosphate cycle with possible interesting applications in the industry comparable to reported enzyme activities for mesophilic fungi. Unfortunately, soil-associated fungi of this region are poorly described and have never been analyzed for their enzymatic hydrolyzation capacities to the best of our knowledge.

## 2. Materials and Methods

### 2.1. Sample Collection and Preparation

Two sampling sites were established, each at two different altitudes within the grassland paramo (4000 m asl) and at the frontier between grassland paramo and superparamo (4150 m asl), lying on a lineal transect towards the peak of the volcano Northern Iliniza to guarantee similar climatic conditions and soil characteristics (Figure 1). Each sampling site was carefully selected for level ground, open vegetation in a good conservation state, and the absence of animal signs or human interference. Georeferencing with a GPS (Garmin, Olathe, KS, USA) was applied at each sampling site (Appendix A). Then, 10 cm of topsoil was removed together with vegetation, and three soil samples were taken with sterile instruments, giving 12 samples. Data dataloggers (HOBO, Lakeville, MN, USA) were established at different altitudes to collect temperature data. Soil samples were analyzed by the Ecuadorian National Institute for Agricultural Research (INIAP) for organic matter (OM), total nitrogen (Nt), phosphorus, potassium, calcium magnesium (Olsen modified), and sulfur (calcium phosphate standard protocols). Conductivity was measured with water-saturated paste and pH in a soil-to-water ratio of 1:2.5.

Soil samples were diluted with sterile water at dilutions of 0.1 g in 10, 100, and 1000 mL with three replicates each, and 50 µL of each dilution were distributed with a sterilized spreading rod on 10 cm Petri dish containing 12 mL of potato dextrose agar (PDA) (Sigma–Aldrich, Saint Louis, MO, USA) with streptomycin at 1 mg/mL (GM, London, UK). Then, Petri dishes were incubated for 48 h at 20 °C. After incubation, eight samples of clearly isolated and visibly round-shaped fungal colonies were picked with sterile toothpicks and transferred to a fresh PDA petri dish for cultivation.

For the growth assay, each fungus was picked with a sterile toothpick and inoculated at the center of a PDA Petri dish. Then dishes were incubated at 4, 30, and 40 °C, respectively, for a total of 35 days. Growth was measured weekly in cm growth-diameter of the halo through the inoculation point.

### 2.2. Fungal Strain Identification

#### 2.2.1. DNA Extraction

To obtain genomic DNA of three fungal strains, the mycelia were macerated with a mortar and pestle in TES lysis buffer (Tris 100 mM; EDTA 10 mM; 2% SDS). First, lysed tissue was incubated at 65 °C for 15 min. Then, 140 µL of 5 M NaCl were added, and the mixture was incubated on ice for 30 min. Afterward, 600 µL of chloroform/isoamyl alcohol (24:1) were added and centrifuged at 10,000× *g* for 10 min at 4 °C. The supernatant was isolated and mixed with 50 µL sodium acetate 3 M (pH 5.2) and 300 µL isopropanol. After the second centrifugation under the same conditions, the supernatant was discarded, and the mixture was washed twice with 600 µL of 70% ethanol following centrifugation steps. After discarding the final supernatant, the resulting pellet was diluted in 50 µL TE buffer (Tris 10 mM; EDTA 1 mM) and 5 µL RNAse (10 mg/mL).

#### 2.2.2. Polymerase Chain Reaction

Genomic DNA was used to amplify the fungal ITS region (ITS1-5.8S-ITS2) applying the primer pairs ITS4 and ITS5 [29]. For amplification reactions, a PCR Master Mix Kit was used (Promega, Madison, WI, USA), following the manufacturer’s instructions. To visualize the amplification, product electrophoresis was performed on a 1% agarose gel stained with Nancy dye (Sigma–Aldrich, Saint Louis, MO, USA). Next, the amplification products were purified using the Wizard^®^ SV Gel kit and PCR Clean-Up System (Promega) following the kit’s instructions. Finally, the PCR product was quantified on a NanoDrop^®^ (Thermo Scientific, Waltham, MA, USA).

#### 2.2.3. DNA Sequencing

Sequencing reactions were performed with the BigDye^®^ Terminator Cycle Sequencing Kit (Life Technologies, Carlsbad, CA, USA) following the manufacturer’s instructions and analyzed with ABI 3500 XL sequencer system (Life Technologies). Resulting forward and reverse sequences were quality checked and merged into a consensus sequence with BioEdit v.7.0.5.3 [30]. The BLASTn tool of the public NCBI-GenBank (www.ncbi.nlm.nih.gov) and the Trichokey database (http://isth.info/) were used to compare contigs with homologous sequences (both accessed on 12 April 2019). After a second, quality control sequences were aligned with homologous sequences from culture collections applying the ClustalW tool [31]. Then, the sequences were subjected to phylogenetic analysis. The phylogeny was assembled using the neighbor-joining method, calculating the evolutionary distance using the 2-parameter Kimura model, and are expressed as the units of the number of base substitutions per site. The rate variation among sites was modeled with a gamma distribution (shape parameter = 1). The analysis involved 58 nucleotide sequences. All positions containing gaps and missing data were eliminated. There was a total of 430 positions in the final data set. Tree support was calculated with bootstrap analysis with 1000 pseudo-replications, and the tree was inferred using MEGA v.7.0 [32].

### 2.3. Preparation of Crude Enzyme Extract

Isolated fungi were pre-cultured at 20 °C on inclined agar PDA medium in test tubes until sporulation. Concentrations of 10^6^–10^7^ spores per mL were established in sterile water using a Neubauer chamber for counting. These spores were then used to inoculate 50 mL of sterile Adam’s medium (pH = 6) with 1% wheat bran in 125 mL Erlenmeyer flasks (Adam’s medium for 50 mL: 0.05 g KH_2_PO_4_, 0.025 g MgSO_4_·7H_2_O, 0.1 g yeast extract, and 1 g glucose + 50 mL H_2_O distilled). Triplicates were incubated at 20 °C under static and shaking conditions (120 rpm), respectively, for 120 h harvesting each 24 h. Each crude extract sample from flasks was filtered over Whatman paper using a vacuum pump and applied on the same day for enzyme assays.

### 2.4. Enzyme Activity Assay

#### 2.4.1. Enzyme Determination with Natural Substrates

To measure enzyme activity, the crude enzyme extract was mixed with respective water-diluted substrates to obtain activity over time according to the Miller method [33]. Reducing sugars were quantified using 3,5-dinitrosalicylic acid (DNS) combined with 0.5% carboxymethylcellulose (CMC) (endoglucanase measurement) (Sigma–Aldrich^®^) and 0.5% xylan beechwood (endoxylanase measurement) in the following relation: 25 µL substrate, 10 µL buffer sodium acetate 50 mM pH 5.0, and 15 µL crude extract. Mixes were incubated in a thermocycler (Eppendorf^®^, Hamburg, Germany) for 20 min at 30 °C. In the assay to determine optimal incubation temperatures, a range was established from 4 to 32 °C. Then, 50 µL DNS was added to interrupt enzyme activity, and sample-absorbance was measured at 540 nm on a spectrophotometer (Shimadzu, Kyoto, Japan) compared to glucose and xylose standard curves (0–1 mg/mL). Blancs were established adding enzyme extract after incubation, directly interrupting the reaction with DNS. The results were expressed as milliunits per mL (mU/mL), defined as the enzyme quantity that releases one µmol of reducing sugars per minute per mL.

#### 2.4.2. Enzyme Determination with Synthetic Substrates

To measure enzyme activity, the crude enzyme extract was mixed with respective water-diluted substrates to obtain activity over time. The amount of released *p*-nitrophenol as a result of enzyme cleavage was measured for the following substrates: *p*-nitrophenol-β-D-glycopyranoside (β-D-glucosidase measurement), *p*-nitrophenol-β-D-xylanopyranoside (β-D-xylosidase measurement), and *p*-nitrophenol-β-D-cellobiose (exoglucanase measurement) (all substrates obtained from Sigma––Aldrich^®^) used in the following relation: 25 µL substrate, 10 µL sodium acetate buffer 50 mM pH 5.0, and 15 µL crude extract [34].

For the measurement of phosphatases with *p-*nitrophenyl phosphate (acid and alkaline phosphatase), the assay mix was prepared accordingly using 10 µL crude extract, 40 µL substrate, and 100 µL buffer, where acid phosphatase was measured with acetate buffer 100 mM, pH 4.5 and alkaline phosphatase with Tris-HCl 100 mM, pH 8.0 (modified according to [35]). Mixes were incubated in a thermocycler (Eppendorf^®^) for 20 min at 30 °C. In the assay to determine optimal incubation temperatures, a range was established from 4 to 32 °C in steps of 4 °C. Then, 50 µL (100 µL for phosphatases) of 0.2 M Na_2_CO_3_ solution was added to interrupt enzyme activity and sample-absorbance was measured at 410 nm compared to a *p*-nitrophenol–standard curve (0–1 mg/mL) [36]. Blancs were established by adding enzyme extract after incubation, directly interrupting the reaction with Na_2_CO_3_ solution. The results are expressed as milliunits per mL (mU/mL), defined as the enzyme quantity that releases one µmol of *p*-nitrophenol per minute per mL.

### 2.5. Statistical Analysis

Measured enzyme activity is expressed as the mean ± standard deviation using Microsoft EXCEL. The Shapiro–Wilk test was used to prove the normal distribution of data and the Levene’s test for homoscedasticity. To test for statistical difference between incubation types (i.e., agitation and static), averages of the maximum enzyme activities were compared using the *t*-test for independent variables or Mann–Whitney U test, respectively. To analyze maximum yields among the three studied fungi, one-way ANOVA or Kruskal–Wallis’s test were applied following Sidak–Bonferroni or Dunn post-hoc tests. All statistical analyses were performed using licensed STATA 16.0.

## 3. Results and Discussion

### 3.1. Sample Site Conditions and Soil Characteristics

This sampling sites in the Iliniza National Reserve were chosen due to the fact of their proximity to glacial areas of stratified volcanos in the High Andean region. At a 4000 m altitude, the vegetation zone of conserved paramo highlands is grassland, mainly characterized by herbaceous plants such as sphagnum mosses, tussock grass, and rosette plants (Figure 1b,c). At 4150 m asl, the zone of the superparamo builds the frontier between the last zone of abundant vegetation and the beginning of the rocky glacial zone with only sporadic vegetation. Close to the equator, the radiation at this altitude is very high, causing drastic daily temperature changes between day and nighttime with an average temperature of 9.6 °C (Appendix A). Due to the volcanic ash deposition and slow decomposition, the slightly acidic soil (pH 5.87–6.15) has mainly a sandy–loam texture with 7.7–9% organic matter, which is very high, as expected for Andisols [37]. The stabilization of large organic particles also accounts for its vast water retention capacity [38], which, together with the related oxygen availability, directly interferes with the development of microorganisms and the solubilization of phosphorus [39]. The formation of humic acids in the breakdown of organic matter allows for the formation of metal–humus complexes retaining chelated ions like calcium [40] and magnesium [41], which might explain their abundance in analyzed soils and further account for the observed high levels of conductivity [42]. On the other hand, under the environmental conditions of the paramo region, organic matter breakdown is slow. It might negatively interfere with nutrient availability due to the microbial activity as indicated by the observed low phosphorus, sulfur [43], and nitrogen levels (Table 1) [44,45,46].

### 3.2. Screening for Cultivable Fungi

The first screening for cultivable fungal strains from soil samples showed relatively few growing colonies clearly separated from each other, allowing for easy isolation of distinct strains. Out of 12 soil samples, eight colonies were chosen for the study. These strains were further screened for their capacity to grow at different temperature conditions. Generally, all strains showed better growth at 4 °C than 30 °C, indicating their adaptation to cold environments (Appendix A). At 40 °C, no development could be detected. The results showed a potentially psychrophilic profile for the best growing behavior of strain 3.3 at 4 °C given that it was not able to grow at 30 °C and psychrotrophic profiles for the strains 1.1 and 3.1 at 30 °C [5], which were chosen for further experimentation (Appendix A).

### 3.3. Identification of Fungi

Phylogenetic analysis of the ITS region revealed that all three fungi were ascomycetes with strains 1.1 and 3.1 belonging to the genus *Cladosporium: C. michoacanense* and *Cladosporium* sp. 3.1 (*C. Cladosporioides* species complex), respectively, and 3.3 to the genus *Didymella* (Figure 2). Since the ITS region is not sufficient for species delimitation in some of these groups, other secondary barcodes would be needed for more accurate identification. The genus *Cladosporium* is a highly heterogeneous group with cosmopolitan distribution and strong capability of adaptation [47] with species formerly isolated from arctic soils [8,23] and marine sponges [48]. A third strain belonged to the genus *Didymella* which has mainly been studied due to the existence of pathological strains and plant–host interaction [49] but so far to a lesser extent due to the fact of its resistance to extreme conditions and, therefore, its arising enzymatic capacities [50]. Both *Cladosporium* and *Didymella* strains have formerly been reported in soil isolates from Himalaya mountain environments [51,52]. Still, only *Cladosporium* has formerly been described for Andean regions [12,14], and the present study is the first registry of the genus *Didymella* in the paramo ecosystem.

Images taken of the growth behavior in culture dishes of PDA indicate a dark brown colorization of mycelia, which possibly shows intense melanin pigmentation as a protective adaptation to the strong radiation in their natural habitat as formerly described [53,54] (Appendix A). Furthermore, differential growth behavior was detected under static and agitation conditions (Appendix A). Especially, the *Cladosporium* strains tended to grow better under agitation, developing a darker aspect apparently due to the stronger sporulation [55]. Meanwhile, its impact on *Didymella* was less evident.

### 3.4. Enzymatic Characterization

The following section shows the incubation time-dependent enzyme secretion results from 24 to 120 h under agitation and static condition for each isolated fungal strain to characterize the hydrolyzation capacities at 30 °C.

#### 3.4.1. Enzymatic Production of *Cladosporium michoacanense*

*Cladosporium michoacanense* 1.1 showed the most potent activation under agitation with a relatively late onset of enzyme secretion compared to studies analyzing the same enzymes [34] (Table 2). The most significant activity was detected at 72 h for the hydrolyzation of CMC, xylan, and *p*np-phosphate in an alkaline medium; 96 h for *p*np-β-D-glycopyranoside; 120 h for *p*np-β-D-cellobiose, *p*np-β-D-xylanopyranoside, and *p*np-phosphate in acidic medium, indicating statistically significant differences to measurements under static conditions for all enzymes except for exoglucanase and β-glucosidase. Interestingly, under static incubation conditions, the most potent enzyme activity was measured early between 24 to 48 h for endoglucanase, xylanase, β-xylosidase, and alkaline phosphatase and late response at 120 h for exoglucanase, β-glucosidase, and acid phosphatase. Overall, the most efficient enzyme activity for endoglucanase and xylanase could be observed at levels well above 2000 mU/mL under agitated cultivation.

#### 3.4.2. Enzymatic production of *Cladosporium* sp. (*C. cladosporioides* Species Complex)

*Cladosporium* sp. 3.1 indicated the best hydrolyzation performance under agitation at 120 h for most enzymes (Table 3). The late onset in enzyme production is consistent with previous studies [56]. Exceptions were endoglucanase with the highest enzyme secretion at 48 h, β-xylosidase at 24 h, and xylanase at 72 h. Under static conditions, the strongest enzyme secretion was observed at 72–96 h for most enzymes with significantly lower activity values than under agitation conditions, which is coherent with the increased pellet formation observed by comparing the incubation flasks of *C. Cladosporioides* under static and agitation conditions (Appendix A). This phenomenon was formerly described by Raikumar et al. for this genus [55]. Xylanase was the only exception that performed better under static conditions at 72 h of incubation. Agitation leads to better aeration and distribution of heat and nutrients bearing growth benefits for many filamentous fungi [57]. Comparing both analyzed *Cladosporium* species, an advanced enzyme production upon agitation was observed, and a relative early specialization in the production of endoglucanase and xylanase indicated a focus on the breakdown of large sugar polymers such as cellulose and hemicellulose [58]. All remaining enzymes showed a rather late onset.

#### 3.4.3. Enzymatic Production of *Didymella* sp.

The *Didymella* strain 3.3 showed a rather heterogeneous behavior (Table 4). Meanwhile, CMC and *p*np-β-D-xylanopyranoside were most effectively hydrolyzed under agitation at 24 h and 48 h, respectively. However, the best xylan hydrolyzation occurred at 72 h and for *p*np-β-D-glycopyranoside, *p*np-β-D-cellobiose, and *p*np-phosphate in alkaline medium at 96 h and 120 h in acid medium. Interestingly, while β-glucosidase, β-xylosidase, acid, and alkaline phosphatase showed low but detectable activity under agitation conditions, no activity could be detected under static conditions. On the other hand, endoglucanase, and xylanase, which generally showed much higher activity, performed considerably stronger under static conditions at 96 h. These results confirm the previously described strong influence that culture conditions can exert over enzyme production [34,59].

#### 3.4.4. Comparison of Relative Enzyme Activity between Acid and Alkaline Phosphatases

Analysis of differential enzyme production between acid and alkaline phosphatase activities generally indicated more substantial secretion of enzymes upon agitation than under static conditions and stronger and prolonged acid phosphatase activity over time. Meanwhile, alkaline phosphatase showed lower and somewhat restricted activation at growth onset for strains 1.1 and 3.1; activation was strongest from 72 to 120 h for strain 3.1 (Appendix A). This behavior matches with the environmental conditions of their habitat, where slightly acidic soil pH was observed (Table 2), which is also the preferred growth condition for most fungi [60]. It further coincides with the tendency that within strain 1.1 and 3.1, measured pH values of crude extract used to drop stronger over time than in strain 3.3 (Appendix A).

#### 3.4.5. Temperature-Dependent Enzyme Activity

After analyzing the best growth time for differential enzyme production, results were used to determine optimal enzyme secretion under different incubation temperatures ranging from 4 to 32 °C under agitation to optimize enzyme production further. The results indicated statistically significant differences among strains (Figure 3). Meanwhile, strain 1.1 showed the best performance at temperatures ranging from 20 to 24 °C for all enzymes related to the carbon cycle and 8–12 °C for phosphatases; strain 3.1 showed somewhat differential activity indicating main enzyme activity for β-glucosidase and exoglucanase from 4 to 8 °C and for endoglucanase, β-xylosidase, and xylanase from 20 to 24 °C. Phosphatases showed opposite behavior with the highest acid phosphatase activity at 12 °C; meanwhile, alkaline phosphatase performed best at 28 °C. This controversial behavior might best be explained by the adaptation capacity of different *Cladosporium* species to varying ecological niches giving rise to its worldwide abundance [47]. It is typical for fungal strains adapted to changing environmental conditions within a single habitat [61,62]. Strain 3.3 finally showed maximum activity for β-glucosidase, endoglucanase, and xylanase at 8 °C and for exoglucanase and β-xylosidase at 20–24 °C. Phosphatase activity remained low, with the highest activity at 16 °C for both acid and alkaline phosphatase.

The combination of optimal incubation time, agitation, and optimal substrate incubation temperature led to several enzymes’ considerably higher activity (Table 5). *Cladosporium* strain 1.1 specialized in producing endoglucanase and xylanase with 4.56 and 3.03 U/mL activities, respectively, both showing significantly higher performances than the other two strains. Comparing these values with formerly described enzymatic activities, even in the mesophilic fungi screenings, especially endoglucanase values, they are quite competitive. For instance, Grujic et al. (2019) [63] described maximum values of 1.1 U/mL for endoglucanase in *Trichoderma guizhouense*, Ghazali et al. (2019) [64] reported the highest values of 3.93 U/mL for *Trichoderma asperellum* and 5.6 U/mL for *Aspergillus niger.* Unfortunately, most studies for psychrotrophic fungi do not apply optimization assays, so comparability is limited. However, considering the advantage of growing under extreme conditions, the *Cladosporium michoacanense* strain is an interesting candidate for potential industrial application under harsh conditions. Moreover, the relatively early onset of enzyme secretion, indicating efficient substrate hydrolyzation at 72 h, represents an important advantage confirming previous results from studies with *Cladosporium* strains [58,65].

On the other hand, *Cladosporium* sp. 3.1 did not reach values as high as strain 1.1 but managed to perform well for all selected enzymes with values above 1 U/mL except alkaline phosphatase. However, the onset of enzyme production appeared relatively late. Interestingly, strain 3.1 showed the significantly highest activation of 2.04 U/mL for exoglucanase at low temperatures, although former studies indicated a relatively poor performance of paramo-derived fungi for these enzymes [11]. Comparing the mesophilic fungi, results matched with reported highest activities of 1.08 U/mL and 2.37 U/mL in *Trichoderma asperellum* and *Aspergillus niger*, respectively. It is also important to mention that this high activity was reached under conditions of only 4 °C. Furthermore, β-glucosidase, which works synergistically with exoglucanase, also performed well at low temperatures (8 °C). Given that *Cladosporium cladosporioides* had formerly been described to exhibit laccase activity [58,66,67], further studies are needed to determine whether the strain could exhibit lignocellulolytic activity for industrial applications under cold conditions. Moreover, strain 3.1 showed the strongest activation for acid phosphatase, reaching 1.27 U/mL. This result further highlights the importance of applying optimization assays for psychrotolerant enzyme activity given incubation time as well as incubation temperature strongly influenced activity. Therefore, maximum acid phosphatase activity in psychrotrophic fungi of 0.03 U/mL, such as described in a screening study by Gawas-Sakhalkar et al. [68], indicates only a limited comparability to the present results.

Finally, *Didymella* 3.3 performed well only for endoglucanase, β-glucosidase, and xylanase, indicating poor performance for the rest of the enzymes, especially phosphatases, which could be related to the phytopathological behavior of many *Didymella* species allowing for the absorption of nutrients from hosts [49]. Nevertheless, all enzymes that performed better showed the highest values at low temperatures of 8 °C, indicating its potential in producing cold-adapted enzymes [9].

Wide-ranging comparison of psychrotrophic fungi enzyme activity is difficult, given that most screening studies indicate qualitative and semi-quantitative data [7,62,69,70,71] or activity data at a single temperature and incubation time [61], which is not directly comparable to the presented data. This strongly indicates the importance of performing studies with a more profound analysis of maximum achievable enzyme activities under optimized conditions for psychrotrophic and psychrophilic fungi isolated from habitats with extreme conditions.

## 4. Conclusions

The present study directly isolated cold-adapted fungal strains from high Andean Paramo soil, of which three strains were chosen to identify the genus and characterize their enzymatic capacity. Selecting for optimal conditions concerning growth period, incubation conditions, and optimal temperature for enzyme performance, maximum enzyme activities were determined for each strain. *Cladosporium* sp. strain 1.1 showed the most competitive enzyme activity of all analyzed strains for endoglucanase and xylanase at 72 h, and agitation and assay temperatures between 24 and 28 °C, indicating an interesting potential for the industrial application processes where reducing sugars are used such as feed animal, food industry, and fuel production. *Cladosporium* strain 3.1 of the *cladosporioides* complex indicated average performance for a broad spectrum of different enzymes except for exoglucanase, which showed the highest activity. Its good performance of acid phosphatase at 12 °C could be applied in the industry of biofertilization to improve the crop yield of plants grown under cold conditions and in the eutrophication of phosphorus from animal feces due to the consumption of phytate in grains [72]. Finally, the *Didymella* strain 3.3 showed high endoglucanase, β-glucosidase, and xylanase activities at temperatures as low as 8 °C, where most enzymes are inactivated. This strain responded closest to a psychrotrophic profile, which could be specifically interesting in industrial processes such as wastewater management and bioremediation in cold conditions, an aspect that might become increasingly important in an industrialized world tending towards energy reduction.

## Figures and Tables

**Figure 1 microorganisms-10-00282-f001:**
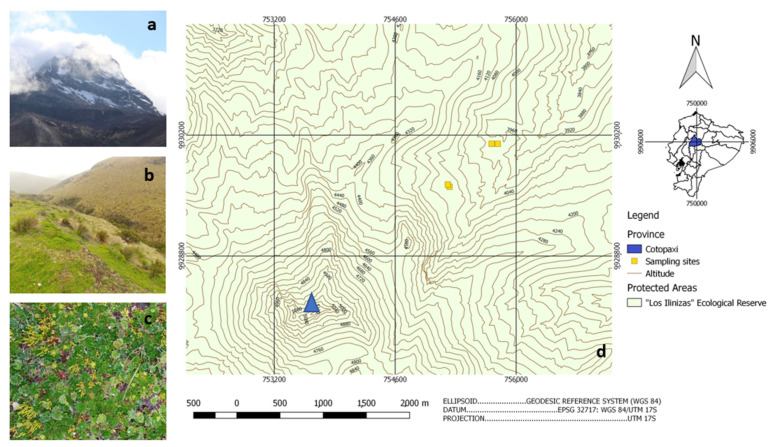
Sampling sites with a vision of the Illiniza north volcano (**a**); typical Andean grass paramo landscape (**b**); typical conserved paramo vegetation (**c**); the georeferenced linear transect of sampling towards the volcano peak (blue triangular) at the bottom left (**d**) (Source: (**a**–**c**) Author 2020; (**d**) modified from [28]).

**Figure 2 microorganisms-10-00282-f002:**
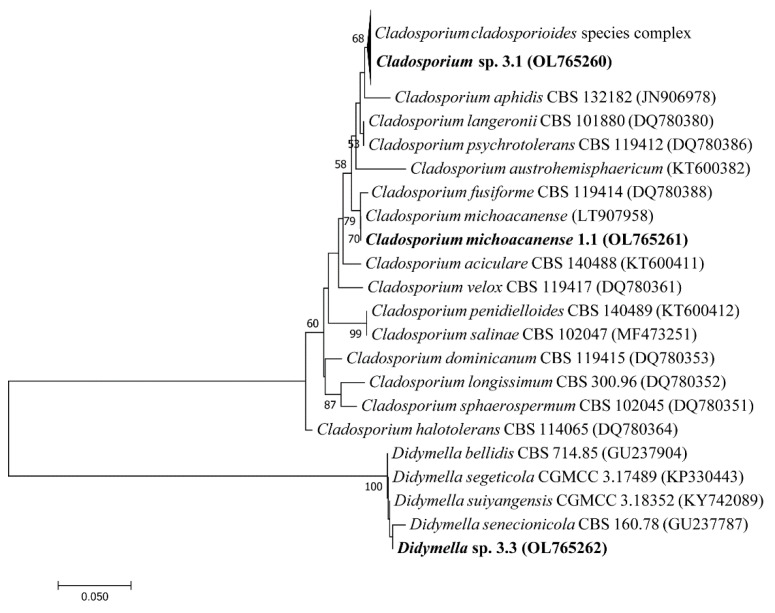
Phylogenetic analysis of filamentous fungi isolated from paramo soil samples (bold) and the closest related species.

**Figure 3 microorganisms-10-00282-f003:**
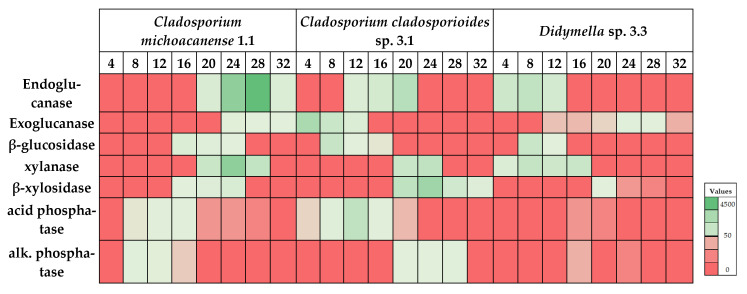
Heatmap analysis of incubation temperature (°C) dependent enzyme performance for each isolated strain. Maximum values ± SD (in mU/mL) are summarized in Table 5.

**Table 1 microorganisms-10-00282-t001:** Soil characteristics at sampling sites.

Sample Site	Soil Type	Soil Texture	pH	COND (µS/cm^2^)	Nt %	OM %	P (ppm)	S (ppm)	K (meq/100 g)	Ca (meq/100 g)	Mg (meq/100 g)
1	Andisol	sandy loam, loamy	6.15	21.5	0.31	8.8	9.01	8.4	0.54	9.78	1.71
2			6.78	21.3	0.36	9.7	14	9.3	0.48	9.76	1.28
3	Andisol	sandy loam	5.93	15.51	0.25	7.7	6.04	4.7	0.42	7.49	1.01
4			5.87	20.8	0.27	9	12	4.1	0.37	4.87	0.73

COND = conductivity, Nt = total nitrogen, and OM = organic matter.

**Table 2 microorganisms-10-00282-t002:** *Cladosporium michoacanense* 1.1 enzymes produced overtime under agitation and static conditions.

	Agitation					Static				
Enzyme	24 h	48 h	72 h	96 h	120 h	24 h	48 h	72 h	96 h	120 h
**Endoglucanase**	932.2 ± 20.7	1514.4 ± 100.5	**2503.7 ± 207.1 ***	394.5 ± 51.8	385.2 ± 170.0	887.4 ± 84.8	**1061.0 ± 23.3**	917.6 ± 17.1	791.4 ± 70.4	730.6 ± 26.9
**Exoglucanase**	ND	ND	ND	1.4 ± 0.7	**1.4 ± 0.3**	ND	ND	ND	ND	**0.2 ± 0.1**
**β-Glucosidase**	ND	ND	ND	**1.7 ± 0.0**	ND	ND	ND	ND	ND	**0.2 ± 0.0**
**Xylanase**	360.2 ± 105.7	969.4 ± 120.1	**2430.1 ± 2.1 ***	274.6 ± 81.3	145.2 ± 34.7	**737.8 ± 88.3**	435.5 ± 111.7	412.2 ± 53.9	353.0 ± 119.5	384.3 ± 70.6
**β-Xylosidase**	ND	ND	0.7±0.	1.1 ± 0.3	**1.4 ± 0.3 ***	**0.5 ± 0.1**	ND	ND	ND	ND
**Acid phosphatase**	1.4 ± 0.6	1.7 ± 1.0	31.0 ± 0.3	5.9	**34.9 ± 0.1 ***	ND	ND	ND	ND	**0.7 ± 0.4**
**Alkaline phosphatase**	0.3±0.1	1.4 ± 0.3	**2.3 ± 0.4 ***	0.2 ± 0.2	ND	**1.4 ± 0.3**	0.6 ± 0.3	ND	ND	ND

Data are expressed as mU/mL ± standard deviation. ND = not detected; ***** indicates a statistically significant difference (*p* < 0.05) between the average enzyme activity of fungi cultivated under agitation vs. static conditions for each analyzed enzyme. In bold, maximum achieved activity.

**Table 3 microorganisms-10-00282-t003:** *Cladosporium cladosporioides* sp. 3.1 enzyme activity over time und agitation and static conditions.

	Agitation					Static				
Enzyme	24 h	48 h	72 h	96 h	120 h	24 h	48 h	72 h	96 h	120 h
**Endoglucanase**	393.2 ± 97.3	**1274.0 ± 102.0**	328.5 ± 68.6	14.0 ± 11.4	14.6 ± 11.9	761.0 ± 70.4	638.8 ± 140.4	945.3 ± 120.6	**1076.4 ± 149.4**	775.0 ± 156.2
**Exoglucanase**	ND	ND	3.6 ± 1.0	6.0 ± 0.3	**12.0 ± 0.6 ***	ND	ND	0.7 ± 0.4	**0.9 ± 0.6**	1.0 ± 0.0
**β-Glucosidase**	0.2 ± 0.2	0.8 ± 0.2	5.7 ± 1.0	8.7 ± 2.9	**11.1 ± 0.8 ***	ND	ND	**0.7 ± 0.4**	ND	ND
**Xylanase**	156.0 ± 8.3	861.0 ± 11.9	**958.9 ± 15.5**	481.0 ± 57.0	ND	522.1 ± 87.6	660.0 ± 56.1	**1113.0 ± 865**	937.3 ± 40.4	898.0 ± 62.1
**β-Xylosidase**	**1.2 ± 0.3**	0.4 ± 0.1	ND	ND	ND	**1.0 ± 0.1**	ND	ND	ND	ND
**Acid phosphatase**	0.7 ± 0.4	1.9 ± 0.7	3.4 ± 0.7	6.8 ± 0.3	**13.9 ± 0.5 ***	ND	ND	0.3 ± 0.2	**1.0 ± 0.7**	0.9 ± 0.1
**Alkaline phosphatase**	ND	ND	1.2 ± 0.8	1.9 ± 1.6	**8.0 ± 1.5 ***	ND	ND	ND	ND	**1.0 ± 0.3**

Data are expressed as mU/mL ± standard deviation. ND = not detected; ***** indicates a statistically significant difference (*p* < 0.05) between the average enzyme activity of fungi cultivated under agitation vs. static conditions for each analyzed enzyme. In bold, maximum achieved activity.

**Table 4 microorganisms-10-00282-t004:** *Didymella* sp. strain 3.3 enzyme activity over time und agitation and static conditions.

	Agitation					Static				
Enzyme	24 h	48 h	72 h	96 h	120 h	24 h	48 h	72 h	96 h	120 h
**Endoglucanase**	**1077.4 ± 6.7**	1037.4 ± 99.0	354.5 ± 6.7	334.4 ± 10.9	119.0 ± 39.4	870.1 ± 99.8	1243.0 ± 108.1	1151.6 ± 189.5	**1413.0 ± 165.7 ***	926.7 ± 15.0
**Exoglucanase**	0.4 ± 0.2	0.7 ± 0.2	1.0 ± 0.3	**1.7 ± 0.4**	1.7 ± 0.4	0.7 ± 0.2	**1.7 ± 0.1**	0.6 ± 0.2	ND	ND
**β-Glucosidase**	ND	ND	ND	**1.6 ± 0.7**	0.5 ± 0.1	ND	ND	ND	ND	ND
**Xylanase**	202.3 ± 15.0	233.0 ± 98.2	**383.0 ± 103.2**	289.2 ± 62.1	ND	873.2 ± 93.7	893.3 ± 125.9	906.0 ± 186.4	**1623.5 ± 196.4 ***	624.7 ± 141.4
**β-Xylosidase**	**1.0 ± 0.0**	0.5 ± 0.1	0.4 ± 0.1	0.1 ± 0.0	ND	ND	ND	ND	ND	ND
**Acid phosphatase**	ND	2.7 ± 0.6	5.7 ± 1.1	7.3 ± 1.1	**7.5 ± 1.5**	ND	ND	ND	ND	**0.3 ± 0.1**
**Alkaline phosphatase**	0.4 ± 0.3	0.4 ± 0.1	1.8 ± 0.1	**2.1 ± 0.5**	ND	ND	ND	ND	ND	ND

Data are expressed as mU/mL ± standard deviation. ND = not detected; ***** indicates a statistically significant difference (*p* < 0.05) between the average enzyme activity of fungi cultivated under agitation vs. static conditions for each analyzed enzyme. In bold, maximum achieved activity.

**Table 5 microorganisms-10-00282-t005:** Maximum enzyme activity yield at optimal assay temperature and incubation time for each isolated strain.

	*Cladosporium Michoacanense* 1.1			*Cladosporium Cladosporioides* Complex 3.1			*Didymella* sp. 3.3		
Enzyme	Maximum Yield ± SD	T opt °C	IT (h)	Maximum Yield ± SD	T opt °C	IT (h)	Maximum Yield ± SD	T opt °C	IT (h)
**Endoglucanase**	**4563** ± 209 ^a^	28	72	1553 ± 330 ^b^	20	48	1247 ± 21 ^b^	8	72
**Exoglucanase**	107 ± 26 ^a^	24	96	**2037** ± 254 ^b^	4	120	127 ± 5 ^a^	24	96
**β-Glucosidase**	303 ± 39 ^a^	16	96	**1013** ± 151 ^b^	8	96	917 ± 12 ^b^	8	120
**Xylanase**	**3036** ± 634 ^a^	24	72	1290 ± 122 ^ab^	24	72	1150 ± 121 ^b^	8	96
**β-Xylosidase**	430 ± 29 ^ab^	24	120	**2457** ± 336 ^a^	24	120	71 ± 5 ^b^	20	48
**Acid phosphatase**	97 ± 12 ^ab^	12	120	**1273** ± 360 ^a^	12	120	17 ± 5 ^b^	16	120
**Alkaline phosphatase**	127 ± 21 ^a^	8	72	**137** ± 37 ^b^	28	120	26 ± 7 ^a^	16	96

Data expressed are as mU/mL ± standard deviation. IT = optimal incubation time; ND = not detected; the letters ^a^ and ^b^ indicate statistically significant differences among enzyme activities comparing the three different fungi (*p* < 0.05) if the letters are not shared between activities of the same enzyme.

## Data Availability

Not applicable.

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
