# Peer review of "Prospection of Psychrotrophic Filamentous Fungi Isolated from the High Andean Paramo Region of Northern Ecuador: Enzymatic Activity and Molecular Identification"

_microorganisms, 2022, doi:10.3390/microorganisms10020282_

Round 1
Reviewer 1 Report
Authors can follow the following suggestions for improvement of manuscript
- Abstract can be revised
- in introduction more literature needed
- at the end of the introduction how present study is important please check again.
- statistical analysis can be elaborated for more accuracy
- Discuaaion of the manuscript can be improve more.
Author Response
Ribeirão Preto, December 16th, 2021.
Dear Reviewer 01, Microorganisms
Find attached the revised version of the manuscript entitled "Prospection of Psychrotrophic Filamentous Fungi Isolated from the High Andean Paramo Region of Northern Ecuador: Enzyme Activity and Molecular Identification" (Microorganisms-1463805). Here, we present the answers to questions and suggestions of reviewer 01. We appreciate the time spent on the revision of this study. We also would like to express our gratitude for the pertinent comments you have made on our work.
We look forward to hearing from you.
Yours sincerely,
Maria de Lourdes Teixeira de Moraes Polizeli
Corresponding author
Response to Reviewer 01:
- Abstract can be revised
A: The abstract has been revised. Errors in the use of italics and the English language have been corrected.
- In Introduction more literature needed
A: Done. We added a breve history of mycological surveys in High Andean paramo regions to the Introduction and added more relevant citations.
- At the end of the introduction how present study is important please check again.
A: Done. We added some aspects to the last paragraph of the introduction.
- Statistical analysis can be elaborated for more accuracy
A: Done. Thank you for the suggestion. We considerably amplified the statistical analysis of the presented data, rewriting Section 2.5 and adding statistical comparison of maximum enzyme activities between fungi cultivation under agitation and under static conditions. The outcomes were added to Tables 2, 3, and 4 of the manuscript. Furthermore, we compared maximum enzyme activity for each enzyme, comparing the performance of each fungal strain. This statistical analysis was added to Table 5. We believe that the newly elaborated statistical analysis improves the overall analysis notably.
- Discussion of the manuscript can be improved more
A: Thank you for the observation. We added extensive parts to the discussion, including a direct comparison of achieved data with other studies, highlighting interesting aspects of measured enzyme activities from isolated fungi. We further pointed out that the study's strength lies in the extensive optimization of quantitative enzyme activity; meanwhile, most studies about psychrotolerant fungi present only qualitative or semi-quantitative data about enzyme activities from isolated fungi under extreme environmental conditions.

Reviewer 2 Report
The paper focuses on a mostly unexplored cold area. I agree with the authors that this may have strong repercussion from a biotechnological point of view. However, I have some concerns about the experimental design which is not so strong. Methods need consistent improvements.
Originality and novelty of the paper is no clear, also because a comprehensive discussion is missing. Data are barely compared with the literature.
In general, please check the English format, because many mistakes are still present.
Please be consistent using italic for genus and species.
total Nitrogen (Nt), phosphorus (P), potassium (K), calcium (Ca) mag-84 nesium (Olsen modified) and sulphur (S): element abbreviations/acronyms can be avoided. We can assume that S is sulphur, no need to explain. Moreover, later in the text, authors continue using for example calcium instead of Ca.
Line 89: hoe many replicas were used for the isolation?
Line 94: why the growth assay was performed at 4-30-40 °C? Why not using something in the middle? Also considering that the isolation was performed at 20 °C.
Fungal identification should be done at species level. Unfortunately, ITS is not the most indicated primer for these fungi. Authors said that ITS region is not sufficient for species delimitation in some of these groups. This is true, but there are recognized alternatives that can be used. For instance, authors should run new PCR run with actin for Cladosporium sp.
Ch. 2.5: this is not a statistical analysis method. Statistical evaluation should be performed.
Line 217: How were these colonies chosen? When you study the biodiversity of a defined environment, mostly if you have few colonies growing, you should isolate all.
Ch. 3.2: data should be better presented. First of all, no fig. S1 is uploaded in the supplementary material.
Authors said that all strains showed better growth. This is not clear from the presented data (at least in Fig. 2). Maybe growth could be compared between the two temperatures providing % of growth? Indeed, Fig. 2 does not report any standard deviation. Here statistical analysis should be performed to defined exactly which are the significant differences.
Who are these strains? Only the code (i.e 3.1) is not sufficient. The identification should be provided. Knowing the species involved may help out the comment of these data and the comparison with the literature. This should be done for all the fungi.
I cannot evaluate some results and comments because Fig. S2-S3-S4 were not uploaded in the supplementary material.
A clear discussion is almost completely missing. All this section should be revised.
Author Response
Dear Reviewer 02, Microorganisms
Find attached the revised version of the manuscript entitled “Prospection of Psychrotrophic Filamentous Fungi Isolated from the High Andean Paramo Region of Northern Ecuador: Enzyme Activity and Molecular Identification" (Microorganisms-1463805). Here, we present the answers to questions and suggestions of reviewer 02. We appreciate the time spent on the revision of this study. We also would like to express our gratitude for the pertinent comments on our work.
We look forward to hearing from you.
Yours sincerely,
Maria de Lourdes Teixeira de Moraes Polizeli
Corresponding author
Response to reviewer 2:
- The paper focuses on a mostly unexplored cold area. I agree with the authors that this may have strong repercussion from a biotechnological point of view. However, I have some concerns about the experimental design which is not so strong. Methods need consistent improvements. Originality and novelty of the paper is no clear, also because a comprehensive discussion is missing. Data are barely compared with the literature.
A: The authors appreciate the revision. We made a considerable effort to remove methodological weak points and add sound statistical analysis and deeper analysis of our results with significant amplification of the discussion to point out the relevance of the presented work. Please find below the detailed description of the applied changes.
More specific comments:
- Please be consistent using italic for genus and species
A: Done. Thank you for the observation. All missing italics were added to the format.
- In general, please check the English format, because many mistakes are still present.
A: Done. English native speakers revised the manuscript to improve the format. Nevertheless, we were unable to identify many committed mistakes. Would you please be so kind so specify a little more? We would be happy to improve the document further.
- Fungal identification should be done at species level. Unfortunately, ITS is not the most indicated primer for these fungi. Authors said that ITS region is not sufficient for species delimitation in some of these groups. This is true, but there are recognized alternatives that can be used. For instance, authors should run new PCR with actin for Cladosporium sp.
We agree with the reviewer that the isolates should be identified at the species level and actin gene, and translation elongation factor 1-α as well, should be sequenced for isolate 3.1. Unfortunately, we do not have all the primer sets (ACT512F/ACT783R and EF728F/EF986R) needed for species-level identification of isolates belonging to the C. cladosporioides complex. However, as the study is more focused on the potential of their enzymes for biotechnological application than on their taxonomy or diversity, we believe that this identification will be enough for now and will not compromise the quality of our study. To overcome this issue, we have improved the discussion by adding more information regarding this genus and its ability to produce enzymes under low temperatures to overcome this issue.
- 2.5: this is not a statistical analysis method. Statistical evaluation should be performed.
A: Done. The whole section was revised and rewritten considering the newly applied methods for statistical analysis in Tables 2, 3, 4, and 5, following the suggestion below.
- Total Nitrogen (Nt), phosphorus (P), potassium (K), calcium (Ca) magnesium (Olsen modified) and sulphur (S): element abbreviations/acronyms can be avoided. We can assume that S is sulphur, no need to explain. Moreover, later in the text, authors continue using for example calcium instead of Ca.
A: Done. Abbreviations for elements were removed.
- Line 89: how many replicas were used for the isolation?
A: Done. The phrase “with three replicates each” was added.
- Line 217: How were these colonies chosen? When you study the biodiversity of a defined environment, mostly if you have few colonies growing, you should isolate all.
A: Colonies were chosen due to the growing behavior of the colonies. It is important to mention that isolated fungi were transported from Ecuador to Brazil in order to perform analysis. Therefore, it was not possible to isolate all the existing colonies as exportation permission is difficult. Therefore, the focus of the work was less to study the diversity of a defined environment but to study the enzymatic capacity of some promising isolated fungi from this area.
- 3.2: data should be better presented. First of all, no fig. S1 is uploaded in the supplementary material. I cannot evaluate some results and comments because Fig. S2-S3-S4 were not uploaded in the supplementary material.
A: We apologize for the technical mistake that only the supplementary table and the supplementary figures were not uploaded. Together with uploading the revised manuscript, this error will be corrected.
- Line 94: Why the growth assay was performed at 4-30-40°C? Why not using something in the middle? Also considering that the isolation was performed at 20°C
A: When fungi were first isolated from paramo soil in Ecuador, we did not yet know which type of fungi we would expect since literature about soil fungi in the area is very scarce. Therefore, the assay in figure 2 was not performed to comprehensively analyze the growth behavior of the isolated fungi comprehensively but rather to screen for attractive candidates for further enzymatic analysis. In volcanic regions, possible isolated could have included from the cryophilic to thermophilic fungi (due to volcanic activity within the region of study), so we tried to rather screen for relatively extreme temperatures to see which would perform best under these circumstances. We already knew to form the isolation of the fungi at 20°C that they were mainly growing well at this temperature.
- Authors said that all strains showed better growth. This is not clear from the presented data (at least in Fig. 2).
A: Indeed, growth difference can be better observed in Figure S1, which, unfortunately, due to an upload error, hadn't been provided in the first manuscript. We apologize for this mistake. We hope the affirmation is better understood with the added supplementary figure S1.
- A: Maybe growth could be compared between the two temperatures providing % of growth? Indeed, Fig. 2 does not report any standard deviation. Here statistical analysis should be performed to defined exactly which are the significant differences.
A: Thank you for the suggestion. We absolutely agree that an assay designed to analyze the growth capacities of fungi rather than screening for interesting candidates would have needed a stronger methodological foundation. Therefore, we consider that the mistake we committed was to put the figure in the article's leading body in the first place. Nevertheless, we wanted to explain how the mainly analyzed fungi were chosen. Hence, figure 2 was removed from the main text body to supplementary data. Since, unfortunately, the experiment had been performed without triplicates, statistical analyses were performed. In addition, the whole section of statistically sound analysis focuses on the article's main objective, which is the maximum achievable yield of enzyme activity.
- Who are these strains? Only the code (i.e 3.1) is not sufficient. The identification should be provided. Knowing the species involved may help out the comment of these data and the comparison with the literature. This should be done for all the fungi.
A: We agree with the reviewer that identifying the 8 originally isolated fungal strains would have added an interesting detail to the work and its discussion. Unfortunately, the unidentified isolates were contaminated during the pandemic lockdown, except the three identified strains kept with isolated frozen spores. Nevertheless, as the focus mainly lies on the analysis of enzymatic activity, we decided to remove figure 2 as it does not contribute to the main objective of the study.
- Originality and novelty of the paper is no clear, also because a comprehensive discussion is missing. Data are barely compared with the literature. A clear discussion is almost completely missing. All this section should be revised.
A: Thank you for the observation. We added extensive parts to the discussion, including directly comparing achieved data with other studies, highlighting interesting aspects of measured enzyme activities from isolated fungi. We further pointed out that the study's strength lies in the extensive optimization of quantitative enzyme activity; meanwhile, most studies about psychrotolerant fungi present only qualitative or semi-quantitative data about enzyme activities from isolated fungi under extreme environmental conditions.

Reviewer 3 Report
Dear Authors,
I am carefully read your manuscript entitled “Prospection of Psychrotrophic Filamentous Fungi Isolated from the High Andean Paramo Region of Northern Ecuador: Enzymatic Activity and Molecular Identification.” This manuscript included exciting results for the screening of enzyme-producing fungi from Andean paramo for the readers. However, I found several weak points in your manuscript. Therefore, I recommend revising the manuscript to overcome the following topics for publication.
- You should add the short history of the mycological survey in the High Andean paramo region of northern Ecuador in the introduction because we did not know whether your isolates were new records in these areas or not. If those of new in these areas, authors should describe this information in your results.
- Why did you choose 20ËšC, a relatively high temperature for psychrophilic species of fungi to isolate from your material? You should explain the reason to 20ËšC to isolate fungi. If you had the results of other low temperatures such as 4 or 10ËšC, please show your results.
- There is no accession no. of your isolates (C. michoacanense 1.1 and Cladosporium sp. 3.1). You should obtain accession no. of your isolates and add in Figure 3 (Figure legend is also not correct).
Author Response
December 16th, 2021.
Dear Reviewer
Microorganisms
Find attached the revised version of the manuscript entitled "Prospection of Psychrotrophic Filamentous Fungi Isolated from the High Andean Paramo Region of Northern Ecuador: Enzyme Activity and Molecular Identification" (Microorganisms-1463805). We appreciate the time spent by the referees on the revision of this study. We also would like to express our gratitude for the pertinent comments on our work.
We look forward to hearing from you.
Yours sincerely,
Maria de Lourdes Teixeira de Moraes Polizeli
Corresponding author
Response to reviewer 03:
- I am carefully read your manuscript entitled "Prospection of Psychrotrophic Filamentous Fungi Isolated from the High Andean Paramo Region of Northern Ecuador: Enzymatic Activity and Molecular Identification." This manuscript included exciting results for the screening of enzyme-producing fungi from Andean paramo for the readers. However, I found several weak points in your manuscript. Therefore, I recommend revising the manuscript to overcome the following topics for publication.
A: Many thanks for the encouraging comment and the careful revision. We hope we can eliminate the weak points and revised all suggestions, detailed as follows.
- You should add the short history of the mycological survey in the High Andean paramo region of northern Ecuador in the introduction because we did not know whether your isolates were new records in these areas or not. If those of new in these areas, authors should describe this information in your results.
A: Done. A brief description of surveys in South American Paramo in general and Ecuador, in particular, was added to the introduction. Furthermore, we described Cladosporium fungal strains for paramo ecosystems, but Didymella was first reported for the study area. Many thanks for the suggestion.
- Why did you choose 20ËšC, a relatively high temperature for psychrophilic species of fungi to isolate from your material? You should explain the reason to 20ËšC to isolate fungi. If you had the results of other low temperatures such as 4 or 10ËšC, please show your results.
- In the beginning of the study, we did not know what kind of fungi we would expect. The isolation at 20°C to stimulate effective growth is already lower than 28°C-30°C which we usually apply for incubation. Interestingly the paramo region also reaches higher temperatures during intensive radiation periods at noon. It is rather characterized by extreme temperature fluctuations than permanent freezing conditions. Nevertheless, we agree that lower temperatures could have been chosen. Therefore, we applied the second screening where 4°C was also selected.
- There is no accession no. of your isolates ( michoacanense 1.1 and Cladosporium sp. 3.1). You should obtain accession no. of your isolates and add in Figure 3 (Figure legend is also not correct).
A: Done. Thank you for this hint! As suggested, the accession number of the sequences generated in this study was added to figure 3. Also, the correct legend was updated. We apologize for the mistake of copying the wrong legend in the original manuscript. Please check the new figure 3 and the figure legend.

Round 2
Reviewer 2 Report
I thank the authors for the response. I still have some concern about some points.
Comment 4
I’m sorry to hear that we do not have all the primer sets for C. cladosporioides complex, but this is kind of important. In alternative, authors should consider depositing the fungi in a recognize Microbial Collection for biodiversity protection and they may also help with the identification.
Comment 8
I understand that the study of the fungi present in the soil sample was not the topic, but this is still a major issue.
By the way, since you mentioned that the isolated fungi were transported from Ecuador to Brazil to perform analysis, did authors have respected the ABS according to the Nagoya protocol? Ecuador signed it in 2011 even though I do not know the actual working conditions for research studies. In order to protect the biodiversity and respect the microbial richness of any Country, this is an issue that should be definitely addressed before publication.
Comment 11 and 12
Figure S1 is indeed helping but it is not a result. Data should be given in a quantitative way (i.e. colony radium) but since you did not run replicas this is difficult. Am I right? I’m sorry but in this way, this comment and results are not worthy of publication. You may consider repeating this experiment (for the fungi you still have) if you want to present data about the temperature of growth. By the way, I do consider this information important, but it should be presented with a scientific soundness.
Comment 13
I’m sorry to hear that most of the isolated fungi are gone. This is a strong weakness to the paper, mostly because the data about the fungi are still preliminary and additional efforts should be done. You may consider to revise the paper, including only the information about the strains that are still alive and those ones that you have complete studies (identification, growth, enzymes, etc.). I can see that authors say that the focus mainly lies on the analysis of enzymatic activity, but these enzymatic activities should be associated to something more than ‘a fungus’.
Author Response
Comment 4
I’m sorry to hear that we do not have all the primer sets for C. cladosporioides complex, but this is kind of important. In alternative, authors should consider depositing the fungi in a recognize Microbial Collection for biodiversity protection and they may also help with the identification.
Thank you for your concern with the identification on species level and your recommendation about the deposition in a Microbial Collection. We would like to clarify that isolated fungi from our laboratory are generally sent to the Mycological library of the University of Pernambuco, Department of Mycology, Brazil (https://www.ufpe.br/micoteca). This library is official and runs under ISO 9001:2015 standards. Within this facility, fungi are stored and identified on species level. The fungi reported in the present work were also sent to this Mycological collection, petition N°72/19, but unfortunately, due to the pandemic situation, the institute remained closed during the period, following the biosecurity guidelines of our country. We believe that upon returning to normal activities and newly introduced governmental policies, working capacities will be restored. In the medium turn, we might receive the identification, given that all samples were carefully sent for storage.
On the other hand, considering that these tests are time-consuming and given that the study is of exploratory character with our focus laid on the enzyme production capacity, we accredit that the given identification is sufficient principally if we proceed with studies using these isolates. We would like to further emphasize that many recent studies in bioprospection were published with isolates different from species-level, which did not compromise the importance of their work (for example, Contato et al., Microorganismos 2021 https://doi.org/10.3390/microorganisms9030533) or even using only preliminary morphological identification (for example, Peraza-Jiménez, J. Fungi 2022 https://doi.org/10.3390/jof8010022). The same way studies emphasizing diversity only used taxonomy based on a single region, without identification on species level of their isolates (for example, Melo et al. Antonie van Leeuwenhoek 2021 https://doi.org/10.1007/s10482-021-01555-1 ).
Comment 8
I understand that the study of the fungi present in the soil sample was not the topic, but this is still a major issue.
By the way, since you mentioned that the isolated fungi were transported from Ecuador to Brazil to perform analysis, did the authors have respected the ABS according to the Nagoya protocol? Ecuador signed it in 2011 even though I do not know the actual working conditions for research studies. In order to protect the biodiversity and respect the microbial richness of any Country, this is an issue that should be definitely addressed before publication.
We appreciate the comment. We formerly did get permissions for isolation and collection of fungi from protected areas signed by the ministry of environment (MAAE), with help from the institute of biodiversity (INABIO) (Contrato Marco N° MAE-DNB-CM-2016-0045 and renovation in MAAE-DBI-CM-2021-0163), including permissions of access to genetic resources, as well as permissions for exportation (Autorización de exportación científica N° 123-2019-EXP-CM-MIC-DNB/MA) and transport (Acuerdo de transferencia de materiales ATM INABIO-USP-2019). Worthy to mention that these permissions were also the reason why the number of isolates from soil samples had to be reduced in the first place.
Comment 11 and 12
Figure S1 is indeed helping but it is not a result. Data should be given in a quantitative way (i.e. colony radium) but since you did not run replicas this is difficult. Am I right? I’m sorry but in this way, this comment and results are not worthy of publication. You may consider repeating this experiment (for the fungi you still have) if you want to present data about the temperature of growth. By the way, I do consider this information important, but it should be presented with a scientific soundness.
We appreciate your recommendations, but, in our understanding, the figure could collaborate as a first criterium of analysis. Many times, images are reported in the sense of complementary information but are not themselves results. This affirmation is true, for example, in some of our publications in journals with similar impacts like Microorganisms, where images were obtained with scanning electron microscopy. We do agree that the presentation of data requires at least three repetitions in order to be able to present standard deviations the way it was done in tables 2, 3, and 4. We would like to state that quantitative data about growth radius is provided in Figure S2 which is directly related to Figure S1. The experiment was undertaken to select for interesting fungi which were further analyzed and are therefore not directly considered a result of the study itself. That is the reason why all data was passed to the supplementary section. We could consider removing it, but we still think it is important to explain how fungi were chosen in the first place and why we considered the three strains.
Comment 13
I’m sorry to hear that most of the isolated fungi are gone. This is a strong weakness to the paper, mostly because the data about the fungi are still preliminary, and additional efforts should be made. You may consider to revise the paper, including only the information about the strains that are still alive and those ones that you have complete studies (identification, growth, enzymes, etc.). I can see that authors say that the focus mainly lies on the analysis of enzymatic activity, but these enzymatic activities should be associated to something more than ‘a fungus’.
We believe there was an equivocation in terms used in the answers to the present reviewer for which we would like to apologize. We would like to clarify that the isolates were not lost, but rather some suffered contamination given to a technical problem with a fridge where isolates were stored. As formerly mentioned, we had no access to the laboratory during lockdown phases. Contaminated strains can be restored and purified again in a time-consuming process, as we do in the bioprospection work isolating pure strains from the soil. Therefore, we do not accept this problem as a concern of sloppiness, given that in our laboratory, there are more than a thousand isolated fungi strains from several important projects. We understand the difficulty in obtaining public financing, which is why we work seriously with a strong commitment confirmed by a considerable number of publications (http://lattes.cnpq.br/0162059400153020). Furthermore, as mentioned before, all samples were sent to the mycological library in the University of Pernambuco, which is internationally recognized. Once again, we want to emphasize that the principal focus of the study lay on the three strains we consider most interesting. We could consider removing figures S1 and S2, which are a minor part of the study, but still, we feel they are part of the article's history.
- We would like to request the attention of the reviewer and Editor of the journal in this paragraph. We raise the alert that the publication of this manuscript seals the partnership established between the countries Brazil and Ecuador (Convenio académico internacional USP-UCE-2016). Moreover, the information is unprecedented and will lead to the opening of new lines of research in Ecuador by the team of assistant professor Stefan Brück at the Central University of Ecuador. The work could be continued with the focus on the biotechnical application of microorganisms, environmental ecology, taxonomy, biochemistry, and molecular biology. Thus, it will undoubtedly be a manuscript with many citations by the team, which will impact the value of the journal in question. We would also like to remind you that the other two reviewers have already approved the manuscript, showing the present work's credibility. Finally, we most sincerely appreciate all received attention, the suggestions, and the inverted work in the analysis of our work. Thank you very much.

Reviewer 3 Report
Dear Authors,
I am satisfied with your revised manuscript and recommend publishing it in this journal.
Author Response
Reviewer 03, Microorganisms
Dear Authors,
I am satisfied with your revised manuscript and recommend publishing it in this journal.
Dear Reviewer 03
We appreciate the time spent on the revision of this study. We also would like to express our gratitude for the pertinent comments you have made on our work.
Yours sincerely,
Maria de Lourdes Teixeira de Moraes Polizeli, Ph.D.
Corresponding author
